# Dynamics of Mitochondrial Proteome and Acetylome in Glioblastoma Cells with Contrasting Metabolic Phenotypes

**DOI:** 10.3390/ijms25063450

**Published:** 2024-03-19

**Authors:** Diana Lashidua Fernández-Coto, Jeovanis Gil, Guadalupe Ayala, Sergio Encarnación-Guevara

**Affiliations:** 1Laboratorio de Proteómica, Centro de Ciencias Genómicas-UNAM, Universidad s/n, Col. Chamilpa, Cuernavaca 62210, Mexico; cocodi_trent@hotmail.com; 2Department of Translational Medicine, Lund University, 22242 Lund, Sweden; jeovanis.gil_valdes@med.lu.se; 3Instituto Nacional de Salud Pública, Universidad No. 655, Col. Santa María Ahuacatitlán, Cuernavaca 62100, Mexico; gayala@insp.mx

**Keywords:** glioblastoma, lysine acetylation, SIRT3, metabolism, mitochondria

## Abstract

Glioblastoma, a type of cancer affecting the central nervous system, is characterized by its poor prognosis and the dynamic alteration of its metabolic phenotype to fuel development and progression. Critical to cellular metabolism, mitochondria play a pivotal role, where the acetylation of lysine residues on mitochondrial enzymes emerges as a crucial regulatory mechanism of protein function. This post-translational modification, which negatively impacts the mitochondrial proteome’s functionality, is modulated by the enzyme sirtuin 3 (SIRT3). Aiming to elucidate the regulatory role of SIRT3 in mitochondrial metabolism within glioblastoma, we employed high-resolution mass spectrometry to analyze the proteome and acetylome of two glioblastoma cell lines, each exhibiting distinct metabolic behaviors, following the chemical inhibition of SIRT3. Our findings reveal that the protein synthesis machinery, regulated by lysine acetylation, significantly influences the metabolic phenotype of these cells. Moreover, we have shed light on potential novel SIRT3 targets, thereby unveiling new avenues for future investigations. This research highlights the critical function of SIRT3 in mitochondrial metabolism and its broader implications for cellular energetics. It also provides a comparative analysis of the proteome and acetylome across glioblastoma cell lines with opposing metabolic phenotypes.

## 1. Introduction

Glioblastoma multiforme (GBM) represents the zenith of malignancy within brain cancer, characterized by ubiquitous aggression and dismal prognosis. Annually, GBM accounts for 2 to 5 new cases per 100,000 individuals across North America and Europe, displaying a predilection for males over females and a rarity in pediatric populations [1]. The dichotomy of GBM into primary and secondary forms delineates its complexity; secondary GBM, emerging from lower-grade astrocytomas around the age of 45, contrasts with the more prevalent primary GBM, which is noted for its invasiveness and predominance in the elderly [2,3,4]. Despite advances, treatment modalities remain standardized, comprising surgical resection, radiation, and temozolomide chemotherapy [4].

The metabolic versatility of GBM, spanning oxidative to glycolytic phenotypes, underpins its heterogeneity [5,6,7]. This metabolic adaptability is not merely a cellular peculiarity but a determinant of tumor aggressiveness, response to therapy, and overall prognosis. Mainly, glycolytically driven GBM cells exhibit enhanced proliferation, invasion, and chemoresistance, underscoring the urgent need to dissect the underpinnings of metabolic regulation [8]. Central to this metabolic regulation is the acetylation of mitochondrial proteins, a post-translational modification with profound implications for cellular energetics and cancer biology [9,10,11,12]. Despite its recognized importance, the precise contribution of mitochondrial protein acetylation to GBM pathogenesis and the metabolic dichotomy within GBM remains poorly understood. This gap in knowledge presents a critical barrier to the development of targeted therapies and the improvement of patient outcomes.

SIRT3, the primary mitochondrial deacetylase, emerges as a pivotal regulator of mitochondrial function and metabolism. SIRT3 is implicated in regulating mitochondrial protein acetylation and metabolism, targeting various proteins involved in cell proliferation, cell cycle, ATP production, and metabolic pathways such as the tricarboxylic acid cycle (TCA), fatty acid oxidation, oxidative phosphorylation (OXPHOS), and ketogenesis [13]. The nuanced role of SIRT3 in modulating acetylation and thus maintaining mitochondrial functionality underscores its importance, with its effects varying by cellular context. In cancers reliant on oxidative phosphorylation (OXPHOS), SIRT3 may promote growth, whereas it could suppress tumors in glycolytically dependent cancers [14]. SIRT3, a crucial regulator of cellular energy balance, orchestrates ATP levels through various mechanisms. It governs mitochondrial electron transport by deacetylating proteins such as NDUFA9 and succinate dehydrogenase while safeguarding against oxidative stress by targeting ATP synthase (ATP5A) and HSP70 [15]. Moreover, SIRT3 enhances fatty acid oxidation by deacetylating LCAD [16]. Downregulation of SIRT3 leads to electron transfer chain dysfunction and elevated ROS levels. Additionally, SIRT3 activates AceCS2 to initiate the TCA cycle and modulates enzymes like IDH2 to regulate energy production [17].

Therefore, the study of SIRT3’s role in GBM is justified and essential, promising insights into the metabolic intricacies of GBM and revealing potential therapeutic targets. By employing quantitative proteomics and acetylomics at both cellular and mitochondrial levels in GBM cell lines with distinct metabolic profiles, this study aims to elucidate the regulatory landscape of SIRT3. Through comparative analysis of the T98G (oxidative) and U87MG (glycolytic) cell lines, treated and untreated with a SIRT3 inhibitor, we aim to uncover the differential impacts of SIRT3-mediated acetylation on mitochondrial metabolism. Our findings not only validate the crucial role of SIRT3 across varying metabolic phenotypes but also offer novel insights into the proteomic and acetylomic dynamics underlying GBM’s metabolic flexibility. Integrating cellular and mitochondrial proteomics and acetylomics, this approach provides a comprehensive understanding of the regulatory mechanisms at play. By dissecting the nuances of SIRT3’s influence on GBM metabolism, this study contributes significantly to the escalating field of metabolic oncology, offering new avenues for therapeutic intervention and a deeper understanding of GBM’s molecular underpinnings.

## 2. Results

To establish the metabolic baseline of the two glioblastoma cell lines under investigation, we initially confirmed their distinct metabolic phenotypes: U87MG cells exhibiting a glycolytic phenotype and T98G cells demonstrating an oxidative metabolic phenotype. Oxygen consumption and lactate production assays served as the primary metrics for this characterization. In agreement with their respective metabolic identities, T98G cells showed significantly higher oxygen consumption rates (Figure 1A), indicative of enhanced oxidative metabolism. In contrast, U87MG cells displayed elevated lactate production (Figure 1B), characteristic of glycolytic metabolism. These findings were statistically validated through *t*-tests (oxygen consumption: t = 5.33, *p* = 0.033; lactate production: t = 4.35, *p* = 0.012), underscoring the robustness of the metabolic phenotyping. This confirmation was crucial for understanding the metabolic context in which SIRT3 operates within these cells.

Following the metabolic profiling, cell lines, and their isolated mitochondria were treated with dimethyl sulfoxide (control) or 3-TYP (treated), a selective SIRT3 inhibitor. This preparatory step set the stage for a comprehensive exploration of the proteomic and acetylation landscape influenced by SIRT3 activity within distinct metabolic contexts. We employed quantitative proteomics and acetylomics to dissect the proteome and acetylome of the U87MG and T98G cell lines, alongside their mitochondrial counterparts, under control and SIRT3-inhibited conditions. We conducted this analysis across three biological replicates for each condition using a mass-spectrometry-based approach described in previous studies [18,19,20]. The overarching study design is depicted in Figure 1C, illustrating the approach taken to elucidate the role of SIRT3 in these cellular environments.

The mass spectra generated within the study identified 5992 proteins, among which 1489 were acetylated in 2800 acetylation sites. This dataset reaffirmed the notion that protein abundance represents a limiting factor for detecting acetylation sites [18,19], a relationship visually represented in the abundance distribution histograms for all acetylated proteins identified (Figure 1D). Using the Protein Atlas repository’s annotation of 1119 proteins with confirmed mitochondrial subcellular locations as a benchmark [20,21], our proteomic analysis successfully identified 525 mitochondrial proteins in the global proteome and 547 in the mitochondria-enriched samples (Figure 1E).

Mitochondrial subcellular enrichment was quantitatively assessed, revealing an expected predominance of mitochondrial proteins in the mitochondria-enriched extracts compared to cellular and nuclear proteins (Figure 1F). Principal component analyses underscored the distinct proteome abundance profiles between the two cell lines and treatment conditions, affirming the method’s efficacy in capturing the nuanced differences across sample types (Figure 1G).

In subsequent sections, we delve deeper into the differential impacts of SIRT3 inhibition on the proteomic and acetylation profiles of these metabolically distinct glioblastoma cell lines, unveiling insights into the mitochondrial and cellular response mechanisms.

### 2.1. Proteome Dynamics Underpinning Glioblastoma Cell Phenotypes

To explore dysregulated metabolic phenotypic differences between glioblastoma cell lines at the functional level, we delved into the proteome dynamics, employing functional annotation enrichment analysis on significantly dysregulated proteins. This approach uncovered 2300 proteins with markedly different expressions between the cell lines (*t*-test, FDR 1%), highlighting the molecular underpinnings of their metabolic phenotypes (Figure 2A). The analysis highlighted the aggressive glycolytic phenotype of U87MG cells, characterized by the upregulation of pathways related to synthesis, proliferation, cellular translation, ribosome biogenesis, and RNA splicing. Notably, these cells exhibited an adaptive response to hypoxia, even under normal oxygen level conditions, underscoring their adaptation to meet elevated energy demands. A notable downregulation of immune system-related processes further emphasized the distinctive aggressiveness of the glycolytic phenotype in glioblastomas. Conversely, the oxidative phenotype of T98G cells was characterized by a pronounced upregulation of mitochondrial-related functions, including metabolism, organization, biogenesis, transport, and translation alongside an increase in processes related to the extracellular matrix organization and immune system responses. Differential abundance of proteins directly involved in glycolysis and oxidative phosphorylation provided further evidence of the metabolic delineation of these cells (Figure 2B, Appendix A). These proteomic insights at the functional molecular level shed light on the differences in disease presentation associated with the metabolic mechanisms of tumor cells, offering a clearer understanding of their functional molecular landscape [20,22]. Gene set enrichment analysis (GSEA) of the proteome dynamics confirmed these findings, with oxidative phenotype cells showcasing enrichment of pathways linked to mitochondrial energy metabolism, extracellular matrix organization, and immune system response signaling. Conversely, glycolytic cells were distinguished by the upregulation of synthesis-related pathways, reinforcing the metabolic dichotomy observed (Figure 2C).

### 2.2. Acetylation Landscape Distinguishing Oxidative and Glycolytic Glioblastoma Cell Lines

To unravel the complexities of glioblastoma at the proteome scale, we investigated the acetylation profiles of both oxidative and glycolytic glioblastoma cell lines. Our acetylome analysis employed a method that chemically acetylates lysine residues across the proteome, introducing an acetyl group carrying three deuterium atoms instead of hydrogen [18,23,24]. This incorporation results in a mass shift between the chemically induced and the naturally occurring acetylation, facilitating the precise identification of acetylation sites and the calculation of site-specific acetylation occupancy. Using this approach, we identified 2800 acetylation sites across 1489 proteins, underscoring the widespread impact of acetylation as a regulatory mechanism across various cellular functions [9,23,25,26]. Interestingly, the subcellular distribution of acetylated proteins between the two cell lines showed no marked differences, revealing a nuanced layer of complexity in cellular regulation (Figure 3A). Delving deeper, our findings revealed 1398 acetylated sites in 1058 proteins for the T98G line and 1380 sites in 1049 proteins for the U87MG line. The analysis uncovered 702 acetylation sites unique to T98G and 693 unique to U87MG, with 356 acetylation sites shared, painting a picture of the distinct acetylation landscapes that underpin each cell line’s metabolic phenotype. A closer examination of acetylation stoichiometry revealed significant differences correlating with distinct metabolic functions. The T98G cells exhibited 266 peptides with increased acetylation occupancy (>0.2%), while U87MG showed 272. Pathway enrichment analysis brought these differences into sharper focus, highlighting the specific biological processes influenced by acetylation (Figure 3B).

In the oxidative T98G cell line, the heightened acetylation of signaling pathways components, such as those involved in epidermal growth factor receptor (EGFR) signaling pathways—mitogen-activated protein kinase (MAPK), forkhead box O (FoxO), and the phosphoinositide 3-kinase (PI3K-Akt)—highlights acetylation’s pivotal role in regulating key cellular functions, including proliferation and response to treatment in glioblastoma [27,28]. Similarly, acetylation within the vascular endothelial-derived growth factor (VEGF) signaling pathway underscores its potential influence on angiogenesis [29].

Our analysis also spotlighted acetylated proteins in T98G cells linked to critical brain functions, such as axo-dendritic transport and brain development, as well as pathways associated with gliomas and neurodegeneration. Conversely, the glycolytic U87MG cell line showcased a distinct acetylation profile predominantly involving mitochondrial proteins, implicated in various processes from mRNA degradation to calcium import and mitochondrial integrity. Notably, the glycolytic phenotype was characterized by acetylation patterns indicative of mitochondrial damage, with implications for necroptotic processes and pore complex assembly, emphasizing the mitochondria’s regulatory role in necroptosis through mitochondrial ROS and permeability transition [30]. Furthermore, acetylation in the glycolytic cell line was in proteins that participate in processes related to cancer progression, including those involved in mesenchyme development, actin cytoskeleton organization, and cell junction assembly, pointing to acetylation’s role in facilitating cellular motility, resistance to apoptosis, and ECM transition [31].

Both cell lines displayed enrichment in acetylated proteins belonging to the spliceosome machinery, underscoring the potential regulation of spliceosome activity through acetylation. This modification’s role in splicing may influence the production of proteins that contribute to tumorigenesis [32].

### 2.3. Differential Mitochondrial Proteins Expression following SIRT3 Inhibition in Glioblastoma Cell Lines

To delve into the impact of SIRT3 inhibition on protein expression across glioblastoma cell lines with distinct metabolic profiles, we selectively inhibit SIRT3, employing 3-TYP. This specific inhibitor does not alter SIRT3 expression but effectively curtails its activity [33]. For the T98G cell line treated with 3-TYP, 1554 proteins showed significant upregulation, with pathways related to fatty acid metabolism, the TCA cycle, oxidative phosphorylation, and ribosomal and peroxisomal proteins being particularly enriched. Conversely, proteins associated with glycolysis and the HIF-1 signaling pathway showed a downregulation, along with those involved in the biosynthesis of amino acids and nucleotide sugars and components of the spliceosome and proteasome (Figure 4A).

Mitochondria-enriched samples from the T98G cells revealed an upregulation of 1182 proteins post-treatment, with enhancements in pathways like the TCA cycle, oxidative phosphorylation, fatty acid beta-oxidation, peroxisome, and ribosome biogenesis. Downregulated proteins were linked to the phagosome, spliceosome, endocytosis, focal adhesion, glycolysis, and mitophagy pathways (Figure 4B).

In the U87MG cell line, 3-TYP treatment resulted in the upregulation of 1997 proteins, with functional enrichment analysis highlighting their roles in amino and fatty acid degradation, oxidative phosphorylation, the TCA cycle, peroxisome, and spliceosome. The treatment led to a downregulation of proteins involved in the transport and localization of proteins, biosynthesis of macromolecules, peptides, cellular nitrogen compounds, and translation (Figure 4C). Moreover, the mitochondria-enriched samples from treated U87MG cells showed an increase in 933 proteins involved in mitochondrial complex assembly, oxidative phosphorylation, the TCA cycle, cristae formation, mitochondrial biogenesis, and organization. In contrast, proteins related to apoptosis, cell cycle, Ras pathway, endocytosis, ribosome, peroxisome, and proteasome were found to be downregulated (Figure 4D).

The cellular response to SIRT3 inhibition was distinct across cell lines. Yet, both exhibited overexpression of proteins linked to mitochondrial metabolism, including those related to cristae formation, the Krebs cycle, and the electron transport chain (Figure 4E,F). Contrary to expectations that only the oxidative (mitochondria-dependent) cell line would bolster mitochondrial protein expression following SIRT3 inhibition, both cell lines showed an increase, suggesting an overarching strategy to counteract reduced mitochondrial efficiency. This phenomenon underscores SIRT3’s dual role as a tumor promoter or suppressor, contingent on the cancer context, with implications in various cancer types [34,35,36,37].

Previous research indicated that SIRT3 suppression could disrupt glycolytic metabolism in diabetic mice [38]. Our findings suggest that glioblastoma cells upregulate affected mechanisms to mitigate this impact in response to SIRT3 inhibition, regardless of their primary metabolic pathway. This observation emphasizes the importance of SIRT3 in sustaining both glycolytic and mitochondrial metabolic pathways and, crucially, preserving mitochondrial integrity. Our study highlights SIRT3’s critical role in balancing glycolysis and mitochondrial metabolism, which is pivotal for normal cellular function.

### 2.4. Increased Site-Specific Acetylation of Protein Synthesis Machinery following SIRT3 Inhibition across Metabolic Phenotypes

To gain a deeper understanding of the role of SIRT3 further, we embarked on a functional enrichment analysis focusing on proteins that showed an increase in acetylation occupancy by more than 0.2% following the treatment with the SIRT3 inhibitor, 3TYP in both T98G and U87MG cell lines. The analysis revealed an elevation in site-specific acetylation for 236 sites in the T98G-3TYP cell line and 269 sites in the U87MG-3TYP cell line. When extending the same criterion to mitochondrial-enriched samples, the T98G cell line (Mito-T98G-3TYP) increased site-specific occupancy for 196 acetylation sites. In comparison, the treated U87MG cell line (Mito-U87-3TYP) saw an increase for 183 acetylation sites (Appendix A). We filtered out sites that exhibited an increase in acetylation of more than 0.2%, and we focused only on proteins with mitochondrial localization. Specifically, 40 sites were specific to the oxidative T98G cell line, and 32 were specific to the U87MG cell line, with both cell lines sharing 10 acetylation sites (Appendix A).

The intricate interplay between cellular growth, proliferation, and metabolism, significantly regulated by acetylation, is a focal area of cancer research. Our insights into how SIRT3 inhibition modulates specific pathways and processes offer a refined understanding of its biological ramifications. Figure 5A illustrates the diverse biological functions influenced by acetylation changes post-3TYP treatment, with pie plots detailing acetylation response differences across cellular and mitochondrial contexts.

SIRT3 inhibition notably altered the acetylation profiles of proteins integral to critical signaling pathways, including VEGFR, ALK, and ROBO signaling, affirming acetylation’s regulatory role in cellular signaling. Additionally, alterations in RNA processing and mitochondrial RNA metabolism highlight SIRT3’s broad cellular influence. Intriguingly, proteins essential for nervous system development and axon guidance also experienced acetylation shifts, hinting at SIRT3’s role beyond metabolic regulation. Figure 5B delves into the protein-specific changes induced by SIRT3 inhibition, offering a detailed breakdown. The glycolytic phenotype’s reliance on lactate production through aerobic glycolysis and the strategic regulation of the pyruvate dehydrogenase complex (PDC) by acetylation, particularly the PDHA1 subunit, exemplify SIRT3’s crucial metabolic governance. In oxidative cells, SIRT3’s influence extends to mitochondrial complex V acetylation, which is essential to ATP production optimization. The discovery of SIRT3 targets within the Krebs cycle and intermediary metabolism reinforces its regulatory sweep over energy production in oxidative phenotypes.

Notably, increased acetylation was observed in the protein synthesis machinery, including the ribosomal 55s subunit in oxidative cells, suggesting a nuanced regulation of translation. This, alongside increased acetylation in mitochondrial ribosomal proteins like MRPS5, MRPL16, and MRPS7 in glycolytic cells, indicates a vital role for acetylation in regulating mitochondrial translation machinery across metabolic phenotypes [39]. Furthermore, our study sheds light on regulating proteins like HSPA9 and HSP90AB1 through acetylation, revealing their protective role and impact on cellular stress responses [40,41]. The observation of decreased acetylation in key glycolytic enzymes following SIRT3 inhibition, notably PGK1 and PGAM, highlights the critical role of acetylation in metabolic enzyme activity regulation, pointing to intricate regulatory networks modulating tumor cell metabolism [42,43].

In the context of the oxidative phenotype, the regulatory scope of SIRT3 extends to the acetylation of components within mitochondrial Complex V, underlining the role of deacetylation in optimizing ATP synthesis. Identifying additional SIRT3 targets involved in the Krebs cycle and intermediary metabolism further highlights the enzyme’s broad regulatory influence on energy production mechanisms in oxidative cells. This modulation of critical metabolic pathways via acetylation and deacetylation exemplifies the nuanced regulatory capacity of SIRT3, positioning it as a central mediator in the metabolic reprogramming of tumor cells.

Notably, N-Acetyltransferase 10 (NAT10), the only nuclear protein with lysine acetyltransferase activity, increased acetylation status after the treatment in the glycolytic cell line. This protein has been related to the cell cycle and promotes tumor metastasis [44]. Interestingly, the auto-acetylation of NAT10 at K426 is required to activate rRNA transcription [45]. This is another example of how inhibition of SIRT3 impacts RNA metabolism in our cell lines.

## 3. Discussion

In the realm of glioblastoma research, our manuscript casts a spotlight on the metabolic heterogeneity that characterizes this brain tumor, underpinning the aggressive nature of cells exhibiting a glycolytic phenotype [46]. This heterogeneity complicates treatment strategies and delineates a path for therapeutic innovation. Our comprehensive proteomic and acetylation landscape analysis of glioblastoma cell lines T98G (oxidative) and U87MG (glycolytic) reveals distinct metabolic signatures that align with their tumorigenic capacities [47] and expose potential therapeutic vulnerabilities.

Our findings underscore the pivotal role of SIRT3, a mitochondrial deacetylase, in modulating metabolic enzymes, thus influencing the metabolic orientation of glioblastoma cells. The absence of SIRT3 correlates with a glycolytic shift, offering a targetable axis to manipulate tumor metabolism [48]. SIRT3 activates various metabolic enzymes, thereby enhancing mitochondrial substrate oxidation, ATP production, and oxidative metabolism [49]. Specifically, our proteomic data before SIRT3 inhibition highlighted the enriched processes that sustain tumorigenicity, presenting a dichotomy: the glycolytic U87MG cells thrive on pathways favoring synthesis and proliferation, whereas the oxidative T98G cells harness mitochondrial functions to support their growth.

The therapeutic implications of the metabolic dichotomy observed in glioblastoma cell lines underscore the potential for tailored treatment strategies. Specifically, the glycolytic U87MG cells exhibit an enhanced profile of proteins involved in synthesis, proliferation, and cellular translation, coupled with an activated response to hypoxia, even under normoxic conditions. This contrasts sharply with the oxidative T98G cells, which show a pronounced upregulation of mitochondrial-related functions. Such molecular distinctions offer a roadmap for targeted therapies: glycolytic phenotypes like U87MG could be effectively challenged with glycolysis inhibitors. The notable lactate production by these cells suggests that agents like 2-deoxy-D-glucose (2-DG) could significantly disrupt their energy supply chain, curbing proliferation and increasing their apoptosis risk [50]. Additionally, targeting Hexokinase II, a pivotal enzyme in the glycolytic pathway frequently overexpressed in cancer, with inhibitors such as Lonidamine, might offer another avenue to impede glycolytic energy production [51,52]. Furthermore, the strategic inhibition of HIF-1α, given its role in adapting U87MG cells to hypoxic conditions, could attenuate their aggressive traits [53].

Conversely, glioblastomas exhibiting a mitochondrial-driven phenotype, such as the T98G cells, may benefit from therapies that disrupt mitochondrial metabolism. The dependency of these cells on oxidative phosphorylation and mitochondrial processes suggests that inducing oxidative stress or impairing mitochondrial function with agents like metformin or phenformin could selectively compromise their viability [54,55]. Moreover, leveraging antibiotics that inhibit mitochondrial translation—akin to their action on bacterial ribosomes, such as tetracyclines and macrolides—could offer a novel therapeutic pathway by limiting the metabolic adaptability of these cells [56]. These drugs have shown anticancer potential in various studies and are being explored in clinical trials for their efficacy against different cancer types [57,58,59,60,61,62]. Addressing glioblastomas with an oxidative phenotype by targeting specific mitochondrial enzymes and pathways is a promising strategy. The observed upregulation of mitochondrial functions in T98G cells suggests that inhibitors targeting the TCA cycle or oxidative phosphorylation, such as Complex I inhibitors (e.g., IACS-010759), could effectively disrupt their metabolic machinery [63]. This approach highlights the potential of exploiting the unique metabolic vulnerabilities of glioblastoma cell lines for more effective and personalized therapeutic interventions.

Given the dual challenges posed by inter-individual and intra-tumor heterogeneity and the capacity for metabolic phenotype switching, adopting a combination therapy approach emerges as a strategic avenue for targeting the metabolic adaptability inherent in glioblastomas. A dual inhibition strategy, simultaneously targeting glycolysis and oxidative phosphorylation pathways, holds promise for curtailing glioblastoma’s metabolic versatility. Such a multifaceted approach aims to prevent the tumor cells’ ability to alternate metabolic pathways in response to targeted interventions, thereby potentially enhancing therapeutic efficacy.

The universal upregulation of mitochondrial proteins following SIRT3 inhibition across both cell lines signals a compensatory response to maintain energy production. This underscores the nuanced role of mitochondrial function even in glycolytically dominant glioblastoma cells. This observation opens avenues for targeting the acetylation landscape to modulate tumor metabolism, with implications for signaling pathways critical to tumor progression and cellular homeostasis.

The use of histone deacetylase (HDAC) inhibitors to globally adjust acetylation levels represents a strategic intervention [64]. HDAC inhibitors can potentially mitigate the aggressive tumor phenotypes linked to dysregulated acetylation patterns by affecting a broad spectrum of signaling pathways. This tactic addresses the metabolic aspects of glioblastoma pathophysiology and targets the epigenetic regulators of tumor progression, offering a comprehensive approach to therapy.

However, the true potential of these strategies lies in their integration into a precision medicine framework. Our study highlights the indispensable role of in-depth molecular profiling, including proteomics, in refining patient stratification and treatment customization. By continuously monitoring the proteomic signatures of glioblastomas, we can adapt therapies in real-time, addressing the dynamic nature of tumor evolution and therapeutic resistance [65]. In the context of this study, which delves into the proteomic and acetylation landscapes of glioblastoma cell lines with distinct metabolic phenotypes, the need for in-depth molecular profiling of patient-derived tumor samples becomes particularly salient. Our findings underscore the metabolic heterogeneity inherent in glioblastoma, highlighting how variations in protein acetylation and expression profiles can inform the tumor’s metabolic state and potential vulnerabilities. This emphasizes the crucial role of precision medicine in glioblastoma treatment, where proteomics—alongside other “omics” technologies—can serve as a cornerstone for improving patient stratification, optimizing therapeutic target selection, and identifying companion biomarkers for personalized treatment strategies.

## 4. Materials and Methods

### 4.1. Cell Culture

Human glioblastoma cell lines, T98G and U87MG, were cultured in Eagle’s Minimum Essential Medium (EMEM) (from GIBCO, Grand Island, NY, USA) supplemented with 10% fetal bovine serum and 1% penicillin-streptomycin in 175 cm^2^ sterile culture dishes at 37 °C and 5% CO_2_ atmosphere. The medium was changed regularly, and cells could reach 80% confluence until SIRT3 inhibition treatment and protein extraction.

### 4.2. Oximetry (Contribution of OXPHOS to ATP Consumption)

A YSI model 5360 oxygen meter with a chamber volume of 1.9 mL was utilized. The cell pellet was resuspended in Krebs Ringer buffer (125 mM NaCl, 5 mM KCl, 1 mM MgCl_2_, 1.4 mM CaCl_2_, 1 mM KH_2_PO_4_, 25 mM HEPES, pH 7.4), centrifuged at 1500 rpm for 5 min, and the supernatant was removed. This process was repeated three times altogether to remove any residual medium completely. The final cell pellet was resuspended in 1 mL of Krebs medium and quantified.

The chamber was brought to a volume of 1.9 mL with Krebs medium previously incubated at 37 °C with constant bubbling. The chamber was calibrated to 100% oxygen, added, adding 4 mg of cells.

The recording was allowed to run until a constant slope was observed. Then, 5 µM oligomycin was added until a constant slope was reached. Subsequently, 1 mM CN was added until a constant slope was achieved. Finally, a small amount of dithionite was added to completely reduce any remaining oxygen completely, thus concluding the tracing. Calculation of the ng O_2_/min/mg was performed.

### 4.3. Lactate Measurement (Contribution of Glycolysis to ATP Consumption)

The cell pellet was resuspended in Krebs Ringer buffer (125 mM NaCl, 5 mM KCl, 1 mM MgCl_2_, 1.4 mM CaCl_2_, 1 mM KH_2_PO_4_, 25 mM HEPES, pH 7.4), then centrifuged at 1500 rpm for 5 min, and the supernatant was removed. This process was repeated three times to remove any residual medium completely. The final cell pellet was resuspended in 1 mL of Krebs medium and quantified. 4 mg of cells were taken and brought to a final volume of 2 mL with Krebs medium. This mixture was incubated under agitation at 150 rpm and 37 °C for 10 min.

Two labeled tubes were prepared, one marked as “time 0” and the other as “time 10”. To both tubes, 50 µL of PCA was added. After 10 min of agitation, 500 µL of cells were taken and placed in the tube labeled as “time 0”. 37.5 µL of glucose (200 mM) was added to the remaining cells under agitation, and incubation was continued for an additional 10 min.

At the end of the last 10-min period, 500 µL of cells were taken and placed in the tube labeled as “time 10”. Both tubes were agitated and allowed to rest for 15–30 min at 4 °C. Then, they were centrifuged at 2500 rpm for 5 min. The supernatant was transferred to labeled tubes containing 50 µL of Kodak indicator. 60 µL of a 3 M KOH solution with 0.1 M Tris was added and mixed. Finally, 5 µL increments of this last solution were added until the color turned milky white. The samples were stored at −70 °C for subsequent lactate measurement by spectrophotometry.

As the LDH enzyme contains lactate that maintains its undenatured state, this lactate had to be consumed entirely to avoid interference with the measurements. For this purpose, 20 µL of NAD+ (100 mM), 40 µL of LDH, and 1890 µL of hydrazine lysine medium were placed in 2 mL quartz cells. One cell was used for each sample. The cells were inserted into a diode array computerized spectrophotometer (Agilent 8453 Mill Road, Hampton, NH, USA) calibrated at 340 nm. The spectrophotometer measured absorbance every 3 s, plotted in real-time. Once the lactate in the LDH solution was fully consumed, the absorbances remained constant, and a consistent slope was observed. At this point, 50 µL of previously thawed samples were added at room temperature and centrifuged at 10,000 rpm for 2 min.

The absorbance reading continued every 3 s until the lactate in the samples was completely consumed. The absorbance deltas (Δ) and the nmol/min/mg of glucose consumed by each sample were calculated.

### 4.4. SIRT3 Selective Inhibition Treatment with a 3-TYP Reagent

All the experiments were carried out in triplicate. In each iteration, six 175 cm^2^ culture dishes of every cell line at 80% confluence were treated with the 3-TYP (3-(1H-1,2,3-triazol-4-yl) pyridine) reagent (from Selleck Chem, Houston, TX, USA) at a final concentration of 20 µM in EMEM medium without fetal bovine serum. This treatment was maintained for 24 h for subsequent protein extraction from either mitochondrial or total cellular fractions. Concurrently, six 175 cm^2^ culture dishes for each cell line, also maintained at 80% confluence, were subjected to the same conditions but without applying the 3-TYP reagent, serving as the control sample.

### 4.5. Protein Extraction

For protein extraction from the total cellular fraction in each replicated, cells from six culture dishes of each cell line and condition (with and without treatment) were collected with a scraper in cold Phosphate-buffered saline (PBS) maintained at 4 °C. The collected cells were centrifugated at 208× *g* for five minutes, and the supernatant was removed. Three washes with PBS were performed to remove the remaining medium completely.

The cell count was determined using a Neubauer chamber with trypan blue staining. Six million cells were separated and homogenized in a sodium dodecyl sulfate (SDS) solution (4% SDS, 50 mM Dithiothreitol (DTT), 100 mM Tris-HCl pH 8.6). Pelleted cells underwent incubation in the SDS solution for one minute, followed by sonication on ice. Sonication involved twenty cycles of one minute each, with one-minute rest intervals between cycles. The resulting homogenate was centrifuged at 1500× *g*/10 min/4 °C, and the supernatant was separated for further quantification. Protein content was proved using a 2D Quant kit following the manual instructions (from GE Healthcare, Chicago, IL, USA).

### 4.6. Mitochondrial Purification

To obtain a replicate of the mitochondrial fraction, six million cells from each cell line and condition were homogenized using a glass homogenizer in HEPES solution (20 mM Hepes, 2 mM EGTA, 250 mM sucrose, and 0.5% albumin; pH 7.4). The lysates were centrifuged at 1500× *g*/20 min/4 °C, and the supernatant was collected. This process was repeated three more times, with the supernatants being combined at each step. The resulting mixture was centrifuged at 15,600× *g*/50 min/4 °C. The precipitate obtained was suspended in a mannitol buffer (120 mM mannitol, 70 mM sucrose, 5 mM EDTA, and 5 mM Tris-HCl), which was placed on top of a centrifuge tube containing a sucrose gradient (1.7 M sucrose, 1 M sucrose). The sucrose gradient tube was centrifuged in a swinging rotor at 40,000× *g*/60 min/4 °C. The lower white line contained the mitochondrial phase, which was carefully collected with a micropipette, dissolved in mannitol buffer, and centrifuged at 18,000× *g*/40 min/4° C. The button was homogenized with SDS solution following the procedure in the protein extraction section.

### 4.7. Chemical Acetylation with Deuterated N-Acetoxysuccinimide (NAS-d3)

After obtaining mitochondria extracts and total cellular proteins, we followed the lysine acetylation stoichiometry analysis methodology developed by Jeovanis Gil et al. without any modifications [23]. We applied the in-solution sample preparation procedure as described (SSP).

The protein extracts were incubated at 95 °C for five minutes to reduce the disulfide bridges completely. Iodoacetamide (IAA) was added at a final concentration of 0.1 M, incubated in the dark for 30 min, and centrifuged at 1500× *g* for five minutes. The supernatant was collected and quantified in an SDS-PAGE. 100 µg of each sample was suspended in Triethylammonium bicarbonate (TEAB) solution (100 mM TEAB, 1% SDS, and 0.5% sodium deoxycholate (SDC)). In this methodology, SDS and SDC were essential to avoid protein precipitation. Proteins were precipitated with nine volumes of cold ethanol and incubated overnight at −20 °C, and samples were suspended in 200 µL TEAB.

NAS-d3 alkylating reagent that contains stable heavy isotopes was added to each extract at a ratio of 100:1 (reagent: number of amino groups) and incubated for one hour at room temperature. This reagent is a deuterated N-hydroxysuccinimide (NHS) derivate and acetylates free lysine residues. Proteins suffer fewer collateral reactions than other reagents in residues, such as tyrosine, threonine, and serine.

Then, hydroxylamine was added at a concentration of 5% and incubated for 20 min at room temperature. Immediately after that, proteins were precipitated with ethanol and suspended in a digestion buffer (50 mM albumin; pH 7.5–8, 0.5% SDC). Trypsin was added at a ratio of 1:50 and incubated at 37 °C/16 h. The SDC was eliminated through an ethyl acetate extraction process in an acidic environment. One volume of ethyl acetate was added to the sample and acidified with 0.5% trifluoroacetic acid (TFA). Following thorough vortexing and centrifugation, the organic phase was removed. Another round of ethyl acetate, excluding TFA, was conducted. Lastly, the peptide mixture underwent drying using a SPEED VAC (Thermo Fisher Scientific, Waltham, MA, USA).

The acetylation stoichiometry of all proteins was obtained by mass spectrometry thanks to the distinction in the spectrum of endogenously acetylated proteins and those chemically acetylated with the heaviest reagent.

### 4.8. LC-MS/MS and Data Analysis

We used a mass-spectrometry-based proteomics strategy to describe the protein expression and mapped acetylation sites [23].

Liquid chromatography-tandem mass spectrometry (LC-MS/MS) analysis was performed on a Dionex Ultimate 3000 RSLC nano UPLC system coupled in line with a Q-Exactive high-resolution mass spectrometer (Thermo Fisher Scientific, Waltham, MA, USA). The identification and relative quantification analysis of peptides and proteins were performed with the Max Quant v1.5.3.30 program. The same data generated from LC-MS/MS analysis was used to determine the acetylation stoichiometry using the Pview v2.0 software.

The mass spectrometry proteomics data were submitted to the ProteomeXchange Consortium via the PRIDE [66] partner repository using the data set identifier PXD045197.

Statistical analysis was done with Perseus v1.6.15.0 software [67]. We corrected the intensity data based on the log2 transformation, then the average abundance for each protein was subtracted from the log2 transformation from the individual values. Proteins with fewer than two valid values were filtered out. The corrected data were used to compare the relative quantification of proteins in the three replicates and their controls using the two-sample *t*-test statistical analysis. We obtained relative quantitative information from 6158 proteins. Filtering based on 100% valid values; we performed the principal component analysis (PCA). Functional enrichment analysis of differentially abundant proteins between treated and control samples was performed using Metascape online resource [68], Gene Ontology resource v14 [69], and ToppGene Suite [70]. The reactome pathways, KEGG pathways, and GO biological processes were used to conduct the enrichment analysis. The 2D functional annotation enrichment analysis was conducted using the default parameters with Perseus.

The proteins with acetylation sites with more than 0.2% difference in site occupancy between treated and control samples were submitted to a functional annotation enrichment analysis with Metascape and Cytoscape. The results were plotted with GraphPad Prism v8.0.1 program and RStudio software v R4.2.0.

## 5. Conclusions

Our investigation into the metabolic phenotypes of glioblastoma cell lines through proteomic and acetylation analysis highlights a metabolic dichotomy underscored by distinct acetylation landscapes. The glycolytic phenotype of U87MG cells, characterized by a molecular signature accentuating synthesis and cellular translation, contrasts with the oxidative phenotype of T98G cells, which showcases enhanced mitochondrial functions. This distinction is further supported by our findings on the divergent acetylation patterns: T98G cells exhibit increased acetylation in signaling pathways crucial for cellular proliferation and response to therapies, such as the EGFR, MAPK, FoxO, and PI3K-Akt pathways, along with VEGF signaling that potentially influences angiogenesis. Conversely, U87MG cells show a specific acetylation profile concentrated on mitochondrial proteins, impacting critical processes from mRNA degradation to mitochondrial integrity and suggesting a significant role of acetylation in shaping the glycolytic phenotype.

Crucially, our study delineates the differential roles of SIRT3 across these metabolic profiles. In T98G cells, SIRT3 inhibition enriches pathways related to oxidative metabolism, signifying its indispensable role in supporting oxidative energy production. In contrast, the glycolytic U87MG cells respond to SIRT3 inhibition with an upregulation of proteins that facilitate glycolytic energy production and stress responses, highlighting SIRT3’s flexible regulatory capacity in energy metabolism. Across both cell lines, SIRT3 emerges as a key regulator of mitochondrial integrity, evidenced by the universal overexpression of proteins involved in mitochondrial metabolism following SIRT3 inhibition. This positions SIRT3 as a crucial mediator in maintaining cellular energy balance, irrespective of the metabolic phenotype.

Our investigation into the effects of SIRT3 inhibition on glycolytic U87MG and oxidative T98G glioblastoma cell lines unveils critical differences in their acetylation patterns, shedding light on the nuanced roles of SIRT3 in metabolic regulation. Across both cell lines, an upsurge in acetylation at numerous specific sites—predominantly in mitochondrial proteins—underscores SIRT3’s universal function in fine-tuning cellular metabolism through the precise modulation of protein functionality and interactions. This effect highlights SIRT3’s relevance in controlling mitochondrial proteome acetylation. Notably, oxidative T98G cells demonstrated increased acetylation in mitochondrial ribosomal proteins, essential for sustaining ATP production and supporting oxidative metabolism. Meanwhile, glycolytic U87MG cells responded to SIRT3 inhibition with elevated acetylation levels in proteins linked to stress responses and metabolic adaptability, suggesting a tailored cellular adjustment to the modulation of SIRT3 activity. This differential acetylation points out the adaptive strategies of glioblastoma cells to maintain metabolic efficiency and resilience.

Leveraging advanced mass spectrometry and acetylation stoichiometry techniques, our research sheds light on subtle yet significant acetylation variations that play crucial roles in cellular metabolism. These insights pave the way for further exploration into the therapeutic potential of targeting acetylation and its regulatory enzymes in glioblastoma, aiming to delineate its precise role in tumor metabolism and exploit its regulatory capacity for cancer treatment.

## Figures and Tables

**Figure 1 ijms-25-03450-f001:**
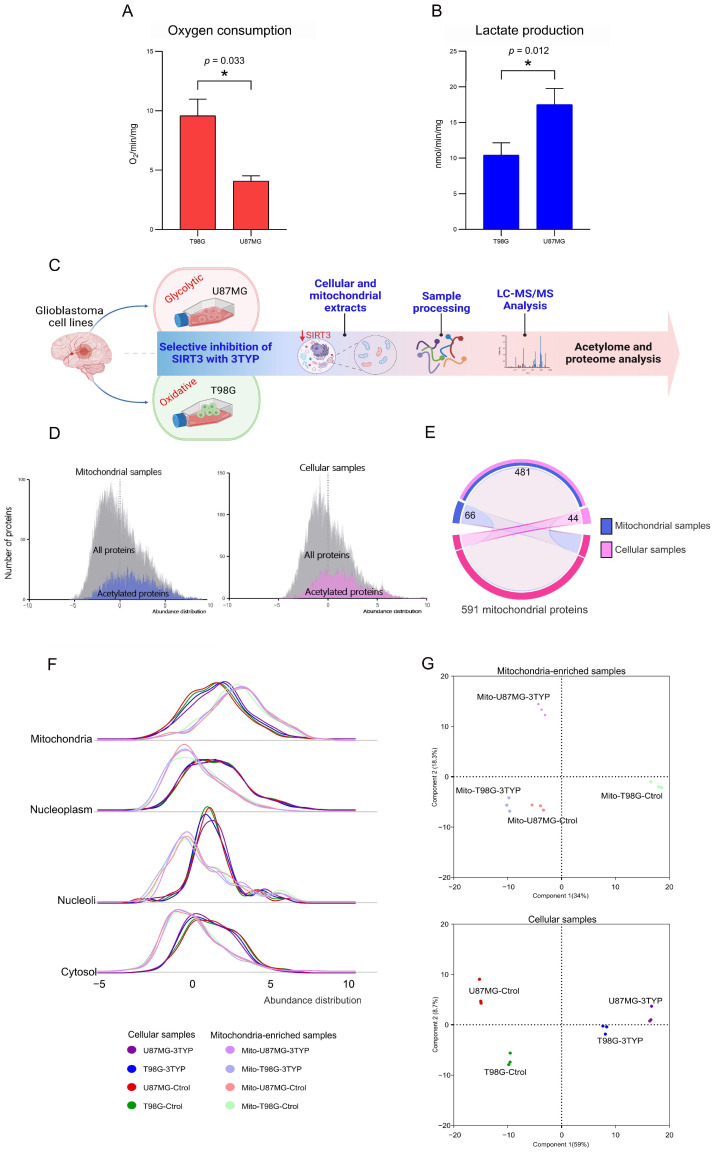
Overview of the molecular phenotyping in glioblastoma cell lines. (**A**,**B**) Oxygen consumption (**A**) and lactate production (**B**) assays confirming the oxidative phenotype of T98G cells (oxygen consumption: t = 5.33, *p* = 0.033) and the glycolytic phenotype of U87MG cells (lactate production: t = 4.35, *p* = 0.012), conducted in triplicate. (**C**) Study design outline illustrating the experimental workflow for the proteome and acetylome characterization under control and SIRT3-inhibited conditions. (**D**) Abundance distribution histograms for all proteins identified, highlighting those with detected acetylation sites. (**E**) Comparative analysis of mitochondrial proteins identified in global proteome and mitochondria-enriched samples. (**F**) Abundance distribution of proteins from each sample based on their cellular location distribution. (**G**) Principal component analyses visualizing the sample distribution according to their proteome profiles. The PCA for mitochondria-enriched samples is shown in the upper plot, while the PCA for whole-cell samples is displayed in the lower plot.

**Figure 2 ijms-25-03450-f002:**
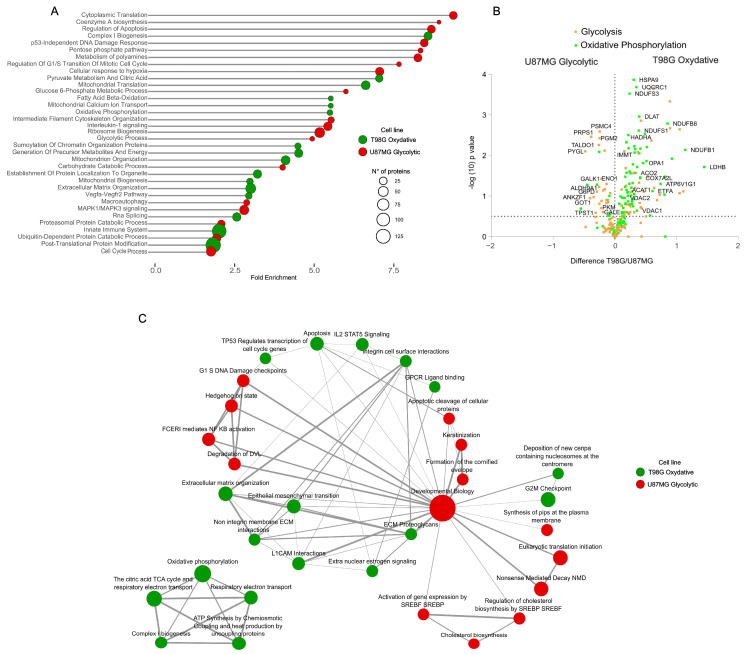
Delineating the functional proteome of glioblastoma cell lines. (**A**) Pathways and processes overrepresented in each cell line, illustrating distinct molecular signatures. (**B**) Differential abundance between cell lines of proteins annotated to be involved in the glycolysis and oxidative phosphorylation, depicted in a volcano plot. (**C**) Gene set enrichment analysis highlights the cell lines’ proteome differences.

**Figure 3 ijms-25-03450-f003:**
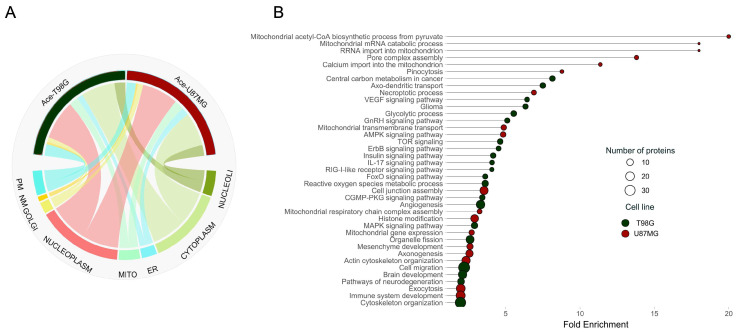
Differentially acetylated proteins highlight distinct pathways in glioblastoma cell lines. (**A**) Chord diagram depicting the distribution of acetylated proteins across various cellular compartments, including nucleoli, cytoplasm, endoplasmic reticulum (ER), mitochondria, nucleoplasm, Golgi apparatus, nuclear membrane (NM), and plasma membrane (PM), illustrating the extensive reach of acetylation across both cell lines. (**B**) visualization of the pathways significantly enriched among proteins with differentially acetylated sites between the oxidative T98G and the glycolytic U87MG cell lines. Dark green and red circles represent pathways in T98G and U87MG cells, respectively.

**Figure 4 ijms-25-03450-f004:**
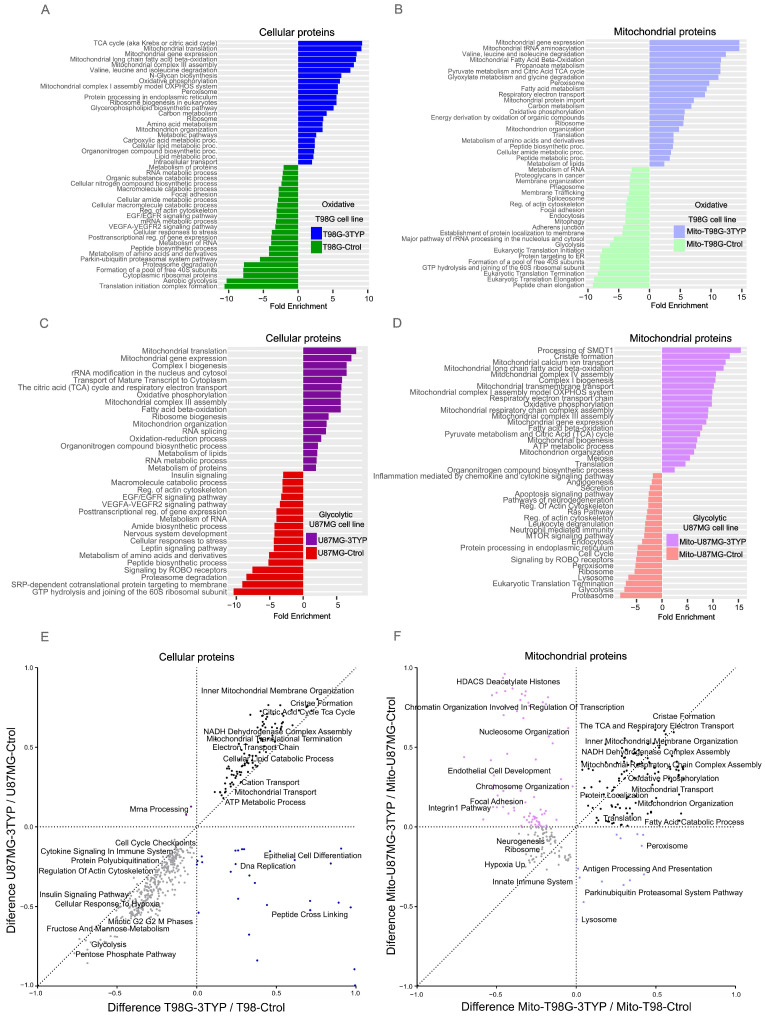
Protein expression profiles following SIRT3 inhibition in glioblastoma cell lines. (**A**) Processes enriched (upregulated) in treated oxidative T98G cells (T98G-3TYP) are indicated in blue, with downregulated processes in green. (**B**) In mitochondrial samples from treated oxidative T98G (Mito-T98G-3TYP), upregulated processes are in light blue and downregulated in light green. (**C**) For the glycolytic U87MG treated cells (U87MG-3TYP), upregulated processes are in purple, with downregulated ones in red. (**D**) Mitochondrial samples from treated glycolytic U87MG (Mito-U87MG-3TYP) show upregulated processes in light purple and downregulated in light red. (**E**,**F**) Display 2D enrichment analysis results, highlighting the unique and shared pathways influenced by SIRT3 inhibition in cellular (**E**) and mitochondrial (**F**) samples.

**Figure 5 ijms-25-03450-f005:**
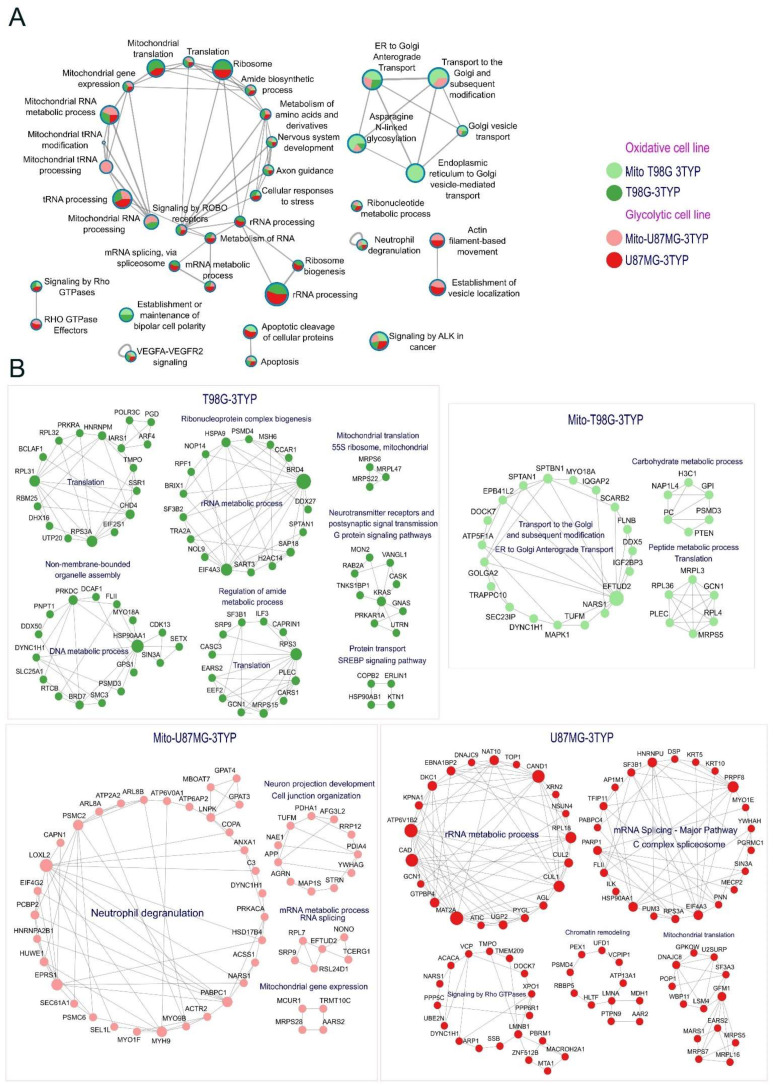
Visualizing the impact of SIRT3 inhibition on protein acetylation. (**A**) Enrichment network of overrepresented biological processes in proteins that increase acetylation after SIRT3 inhibition. Metascape enrichment network, refined in Cytoscape. Nodes appear as pie charts, with the pie size proportional to the Z-Score for specific terms. These charts are color-coded according to the gene list from each sample, where a slice’s size denotes the percentage of genes within the term derived from the respective gene list. The line edge thickness between nodes reflects the similarity SCORE between enriched terms. (**B**) A summary of proteins with increased acetylation post-SIRT3 inhibition is shown for each sample.

## Data Availability

The mass spectrometry proteomics data are available to the ProteomeXchange Consortium via the PRIDE partner repository with the data set identifier PXD045197. Any additional requests can be directed to the corresponding author.

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
