# Peer review of "Dynamics of Mitochondrial Proteome and Acetylome in Glioblastoma Cells with Contrasting Metabolic Phenotypes"

_ijms, 2024, doi:10.3390/ijms25063450_

Round 1

Reviewer 1 Report

Comments and Suggestions for Authors

·      The study did not measure SIRT3 enzymatic activity, which could have provided more insights into the effects of SIRT3 inhibition on protein acetylation.

·      The paper did not investigate the functional consequences of the identified SIRT3 targets, leaving their specific roles and implications for further study.

·      The study focused on two glioblastoma cell lines with different metabolic preferences, which may limit the generalizability of the findings to other cell lines or cancer types.

·      The paper did not explore the long-term effects of SIRT3 inhibition on cellular metabolism and glioblastoma progression, which could provide a more comprehensive understanding of the role of SIRT3 in glioblastoma.

·      The study did not investigate the potential impact of other factors, such as genetic mutations or microenvironmental conditions, on the observed changes in the proteome and acetylome.

·      1What is the purpose of using Max Quant v1.5.3.30 program?

·      What is the significance of using Perseus v1.6.15.0 software for statistical analysis?

·      What are the advantages of using the Metascape online resource for functional enrichment analysis?

·      What are the new SIRT3 targets?

·      What is the impact of SIRT3 inhibition on protein acetylation?

·      How does SIRT3 regulate mitochondrial function?

Comments on the Quality of English Language

Extensive revisions needed. 

Author Response

Reviewer 1

Response to the reviewer comments.

The authors would like to thank the referees and editor for their careful review of our manuscript and for providing us with their comments and suggestions to improve the quality of the manuscript. The following responses have been prepared to address all of the referees’ comments in a point–by–point fashion. 

The revision includes several positive changes based on the review team’s collective input. Based on your guidance, we:

  • Provided more exciting and coherent abstract introduction sections by including the motivation of the study and a more precise description of the specific questions and findings.
  • The text was modified to clarify parts of the manuscript texts, and several sections were included in the reviewers' suggestions.

  • The study did not measure SIRT3 enzymatic activity, which could have provided more insights into the effects of SIRT3 inhibition on protein acetylation

Response: We acknowledge that measuring SIRT3 enzymatic activity could have provided a more comprehensive understanding of the effects of SIRT3 inhibition on protein acetylation. However, we do not consider it essential for our study due to the well-documented inhibitory effect of 3-TYP on SIRT3 activity. In this study, our focus was solely on evaluating the effect of SIRT3 inhibition on protein expression and acetylation in two cell lines with different metabolic preferences. Despite not directly measuring its enzymatic activity, We believe our findings still contribute valuable insights into the role of SIRT3 in protein regulation.

  • The paper did not investigate the functional consequences of the identified SIRT3 targets, leaving their specific roles and implications for further study

Response: We recognize that understanding the specific roles of these targets is critical for contextualizing our findings and advancing the field. However, we believe that addressing this aspect requires a more in-depth study, which we plan to undertake in future research endeavors. In this subsequent study, we intend to delve deeper into the functional implications of these targets, employing a more comprehensive approach to elucidate their specific roles and impact.

  • The study focused on two glioblastoma cell lines with different metabolic preferences, which may limit the generalizability of the findings to other cell lines or cancer types

Response: While our study primarily investigates glioblastoma, it's important to note that our findings extend beyond this specific cancer type. Given that our study explores the effects of SIRT3 inhibition on cellular metabolism, these findings can potentially be extrapolated to other cancer types characterized by both oxidative and glycolytic metabolism. Thus, while our study provides insights into glioblastoma, it also offers a framework for understanding the impact of SIRT3 inhibition on cancers with similar metabolic profiles.

  • The paper did not explore the long-term effects of SIRT3 inhibition on cellular metabolism and glioblastoma progression, which could provide a more comprehensive understanding of the role of SIRT3 in glioblastoma.

Response: Thank you for the reviewer's comment regarding the duration length of inhibition assays in cell lines. Indeed, cell line assays typically have a limited time and focus on observing short-term effects.

However, we acknowledge that fully understanding glioblastoma progression and the role of SIRT3 therein would necessitate a longer-term analysis and models that better reflect the complexity of the disease, such as in vivo models or patient samples at different disease stages.

It is important to note that the cell lines used in our study are convenient models for studying specific molecular mechanisms and cellular phenotypes. While they provide valuable insights into the immediate effects of SIRT3 inhibition, we understand that they cannot fully replicate the heterogeneity and tumor progression observed in glioblastoma patients.

  • The study did not investigate the potential impact of other factors, such as genetic mutations or microenvironmental conditions, on the observed changes in the proteome and acetylome

Response: It is important to note that our study aimed to elucidate the specific molecular mechanisms underlying SIRT3-mediated regulation in glioblastoma cells. However, we acknowledge that the potential impact of other factors, such as genetic mutations or microenvironmental conditions, on the observed changes in the proteome and acetylome warrants further investigation. These factors could play significant roles in influencing cellular responses to SIRT3 modulation. Moving forward, we recognize the importance of incorporating these considerations into more specific and in-depth studies to fully understand the interplay between SIRT3 activity and various cellular contexts. We thank the reviewer for this insightful comment and intend to address these aspects in future research endeavors.

  • What is the purpose of using Max Quant v1.5.3.30 program?

Response: Max Quant is a computational tool. Its primary purpose is to analyze mass spectrometry data obtained from proteomic experiments. Specifically, MaxQuant is designed to perform tasks such as peptide identification and quantification. It utilizes algorithms for protein identification based on comparing experimental mass spectra with theoretical spectra generated from a protein sequence database. MaxQuant facilitates the quantification of peptides and proteins by calculating their abundance levels from the mass spectrometry data.

  • What is the significance of using Perseus v1.6.15.0 software for statistical analysis?

Response: Perseus software is a user-friendly, powerful tool commonly used for statistical analysis in proteomics. Its significance lies in its ability to handle large-scale datasets generated from mass spectrometry experiments and to perform rigorous statistical analysis to identify significant changes in protein abundance or modification across experimental conditions.

Perseus offers a wide range of statistical tools for high-dimensional omics data analysis, including filtering out low-quality data points, removing contaminants, and normalizing data to correct for technical variation and the significance of changes in protein abundance or modification between experimental conditions.

Overall, Perseus is highly significant in proteomics research for its robust statistical analysis capabilities, which enable researchers to derive meaningful insights from complex datasets and advance our understanding of biological processes and disease mechanisms.

  • What are the advantages of using the Metascape online resource for functional enrichment analysis

Response: One of the most essential advantages of using Metascape is its user-friendly interface and easy-to-use tools for data input, analysis, and result visualization. It also integrates data from various public pathway databases, including GO (Gene Ontology), KEGG (Kyoto Encyclopedia of Genes and Genomes), Reactome, and others. Provides interactive visualizations of enrichment results, including bar charts, scatter plots, and network diagrams. Metascape integrates with other bioinformatics tools and databases, allowing users to explore and analyze their results further. For example, it links directly to resources such as NCBI, UniProt, STRING, and Cytoscape.

  • What are the new SIRT3 targets?

Response: The mitochondrial proteins that exhibit increased acetylation upon inhibition of SIRT3 activity can be regarded as potential targets of this enzyme. By referring to supplementary table S2, we identified proteins with an acetylation stoichiometry increase exceeding 0.2 following SIRT3 inhibition. Subsequently, we filtered this dataset to include only mitochondrial proteins and compiled a list of those most likely targeted by SIRT3. This curated list has been included as a supplementary table 1, providing readers easy access to our findings for further consultation. The explanation of this new table was added to the discussion section.

  • What is the impact of SIRT3 inhibition on protein acetylation?

Response: The impact of SIRT3 inhibition on protein acetylation is significant and multifaceted. SIRT3, a mitochondrial deacetylase, is crucial in regulating protein acetylation levels, particularly within the mitochondria. When SIRT3 is inhibited, there is a noticeable increase in protein acetylation across various cellular compartments, not just in the mitochondria. This inhibition alters the acetylation status of numerous proteins involved in critical cellular processes.

Specifically, in our study, SIRT3 inhibition affects proteins associated with signaling pathways, such as vascular endothelial growth factor receptor (VEGFR) signaling, anaplastic lymphoma kinase (ALK) signaling, and ROBO receptor signaling. Additionally, RNA processing and mitochondrial RNA metabolism processes are significantly impacted by SIRT3 inhibition. Moreover, acetylation changes are observed in proteins associated with nervous system development and axon guidance.

Furthermore, the impact of SIRT3 inhibition extends beyond cellular signaling pathways to include metabolic processes. SIRT3 is involved in regulating enzymes critical for energy metabolism, such as those involved in glycolysis, the Krebs cycle, and ATP production. Inhibition of SIRT3 leads to alterations in the acetylation status of these metabolic enzymes, thereby influencing cellular metabolism.

  • How does SIRT3 regulate mitochondrial function?

Response: SIRT3 regulates mitochondrial function as a mitochondrial deacetylase, impacting various aspects of mitochondrial biology.

It promotes mitochondrial function by deacetylating and activating several enzymes involved in energy metabolism, including those of the tricarboxylic acid (TCA) cycle and oxidative phosphorylation (OXPHOS). By activating these enzymes, SIRT3 enhances ATP production and improves cellular respiration.

It influences mitochondrial dynamics by modulating the acetylation status of proteins involved in mitochondrial fusion and fission processes.

It helps mitigate oxidative stress within mitochondria by deacetylating and activating antioxidant enzymes.

It regulates apoptosis (programmed cell death) by modulating the acetylation status of pro-apoptotic and anti-apoptotic proteins within mitochondria. SIRT3 can inhibit apoptosis and promote cell survival under stress conditions by deacetylating certain pro-apoptotic proteins.

It regulates mitochondrial biogenesis by deacetylating transcription factors such as PGC-1α (peroxisome proliferator-activated receptor gamma coactivator 1-alpha), which promotes the expression of genes involved in mitochondrial biogenesis and function.

Reviewer 2 Report

Comments and Suggestions for Authors

The study provides proteomics and acetylation profiles of oxidative and glycolytic glioblastoma cell lines and their mitochondria. Despite the important topic and the experimental part being well performed, visuals are qualitative and well-described, the Manuscript's main hypothesis has presentation gaps and inconsistencies and requires serious revision. Besides, the MS has no Discussion section, and the conclusions are not defined.

Abstract

The abstract lacks consistency. Despite being short, it should provide a continuous story, not a set of unrelated sentences.

Introduction

The research rationale should be elaborated better in the Introduction. There is insufficient justification for investigating mitochondrial protein acetylation with regard to glioblastoma development. It is not enough to say that it "regulates increasing mitochondrial proteins and pathways "(line 56) and that "acetylation of mitochondrial proteins has been found to have a detrimental role on their functions ". There should be a clear indication that mitochondrial protein acetylation can be related to glioblastoma pathogenesis or heterogeneity of the tumour, that protein acetylation is somehow linked to glioblastoma and glioblastoma cell phenotypes, and that SIRT3 has a specific mitochondria-related role in glioblastoma.

The reasoning for investigating cell vs mitochondria proteomics and acetylomics should also be explained. I mean, of course, it is informative, but where is the rationale of this approach? Everything in the Intro is about mitochondrial protein acetylation.

The sentence in line 64 is unclear.

Results

The oxidative vs glycolytic cell profile experiments must be described separately, not in the middle of proteomics/acetylomics analysis, as this is a different methodology and experimental setup. Thus, it should be presented in a consequential way: metabolic profile confirmation first and proteomics/acetylomics – next.

The paragraph's introductory statement: "The regulation of acetylation modifications by SIRT3 is essential for preserving mitochondrial function", lines 218-219, was never backed up with clear examples before in the Introduction. Neither is it explained here. Somewhere in the MS, the role of acetylation and SIRT3 in mitochondrial function must be explained (preferably in the Introduction).

Line 256: "It was unexpected to observe this outcome 256 from the inhibition of SIRT3, given the enzyme's complex role in tumor biology. "– unclear sentence

Line 263: "In a previous study where SIRT3 inhibition affected the glycolytic metabolism of a 263 cell,.. – which cell?

263-271 – it is not clear where the authors are referring to their specific glioblastoma cell type, where some cells are from other studies, and where "cell " in general.

Figure legends are indistinguishable from the main text.

The title of the chapter 2.4. should be more specific. The current version is not informative, as it is well-known evidence that SIRT3 regulates the synthesis of proteins.

Materials and Methods

The subsections are not numbered.

The tense must be kept consistent throughout the Manuscript but not vary from present to past (lines 253-254).

Discussion?

The MS lacks a Discussion section.

Conclusions

The section is, in part, presenting the content that could be fit for discussion. There are no clear conclusions stated.

Considering the claim about mitochondrial protein acetylation significance in glioblastoma tumours, the discussion of this aspect is insufficient.

The conclusions are not defined; the study's main finding is unclear.

Comments on the Quality of English Language

The MS has to be revised for clarity and consistency

Author Response

Reviewer 2

We deeply thank the reviewer for his suggestions regarding the manuscript. We consider that thanks to his guidance, the manuscript we now present has been considerably improved considerably in all its sections.

In this new version, we have tried to present the manuscript with greater clarity, making several revisions in its writing, presentation, and discussion of the data. We hope the reviewer considers it more appropriate in the writing.

Regarding the updating of the literature, we have updated it practically in all parts of the review. Additionally, and thanks to the reviewer's suggestion, we made a new search for information to complement it. This information and new description are now included in the new manuscript version.

Abstract

  • The abstract lacks consistency. Despite being short, it should provide a continuous story, not a set of unrelated sentences.

Response: Firstly, we thank reviewer 2 for their observations on our abstract. You can read the abstract in the new version of the manuscript.

Glioblastoma, an aggressive cancer affecting the central nervous system, undergoes significant metabolic changes to fuel its progression. A crucial regulatory mechanism within cellular metabolism involves the acetylation of lysine residues on mitochondrial and cellular proteins, altering their function. Particularly in mitochondrial proteins, this modification can impair their functionality. However, the mitochondrial enzyme SIRT3 counteracts this acetylation, preserving the integrity of the mitochondrial proteome. To delve into the role of SIRT3 in metabolism, we inhibited SIRT3. We conducted an analysis of the proteome and acetylome using high-resolution mass spectrometry in two distinct glioblastoma cell lines with varying metabolic profiles. Our investigation revealed that the regulation of lysine acetylation influences the protein synthesis machinery, thereby impacting the metabolic phenotype. Additionally, our study identified previously unknown targets of SIRT3, suggesting potential avenues for therapeutic intervention. This research sheds light on the critical role played by SIRT3 in mitochondrial function and its consequential effects on cellular metabolism. Our study of proteome and acetylome dynamics in glioblastoma cell lines offers valuable insights into potential biomarker discovery at both molecular levels.

Introduction

  • The research rationale should be elaborated better in the Introduction. There is insufficient justification for investigating mitochondrial protein acetylation with regard to glioblastoma development. It is not enough to say that it "regulates increasing mitochondrial proteins and pathways "(line 56) and that "acetylation of mitochondrial proteins has been found to have a detrimental role on their functions ". There should be a clear indication that mitochondrial protein acetylation can be related to glioblastoma pathogenesis or heterogeneity of the tumour, that protein acetylation is somehow linked to glioblastoma and glioblastoma cell phenotypes, and that SIRT3 has a specific mitochondria-related role in glioblastoma.
  • The reasoning for investigating cell vs mitochondria proteomics and acetylomics should also be explained. I mean, of course, it is informative, but where is the rationale of this approach? Everything in the Intro is about mitochondrial protein acetylation.
  • The sentence in line 64 is unclear.

Response: Dear Referee, thank you for your feedback. We appreciate the opportunity to clarify and expand upon the research rationale in the Introduction section of our manuscript. We have significantly revised the introduction, emphasizing the unfavorable nature of glycolytic metabolism in glioblastoma. However, despite this understanding, direct evidence of the role of acetylation in the disease's progression or severity remains elusive. This knowledge gap underscores the relevance of our contribution. Specifically, our work focuses on delineating the acetylation profile in cell lines exhibiting varying degrees of severity based on their metabolic characteristics. This aspect of our study is what imbues our work with added value. Due to your comment, We have enriched line 56 and clarified the sentence in line 64.

Results

  • The oxidative vs glycolytic cell profile experiments must be described separately, not in the middle of proteomics/acetylomics analysis, as this is a different methodology and experimental setup. Thus, it should be presented in a consequential way: metabolic profile confirmation first and proteomics/acetylomics – next.

Response: We would like to clarify that the measurement of the metabolic profile was conducted as part of the confirmation process and not as a primary component of our study. We believe that its detailed inclusion in the main body of the text might occupy valuable space without significantly adding to the core results of our research.

For this reason, we chose to include comprehensive details of the experiments related to the metabolic profile in the appendices for those readers interested in examining them in depth.

  • The paragraph's introductory statement: "The regulation of acetylation modifications by SIRT3 is essential for preserving mitochondrial function", lines 218-219, was never backed up with clear examples before in the Introduction. Neither is it explained here. Somewhere in the MS, the role of acetylation and SIRT3 in mitochondrial function must be explained (preferably in the Introduction).

Response: We appreciate your suggestion. In response to your feedback, we have revisited the introduction and have made significant changes to address the concerns raised.

Specifically, we have restructured the introductory statement regarding the regulation of acetylation modifications by SIRT3 to provide more explicit examples and explanations within the introduction. By doing so, we aim to establish a stronger foundation for understanding the role of acetylation and SIRT3 in mitochondrial function from the outset of the manuscript.

We believe that this modification will enhance the overall coherence of our manuscript and provide readers with a more comprehensive understanding of the key concepts discussed.

  • Line 256: "It was unexpected to observe this outcome 256 from the inhibition of SIRT3, given the enzyme's complex role in tumor biology. "– unclear sentence

Response: We have rephrased line 256 to enhance clarity and provide a more detailed explanation: We expected only the mitochondria-dependent cell line (oxidative) to respond by increasing the expression of mitochondrial proteins after inhibiting SIRT3. This expectation was based on the fact that, relying heavily on mitochondrial function, inhibiting this enzyme would lead to a decrease in mitochondrial efficiency. In response, the oxidative cell line would attempt to compensate for this deficiency by increasing the expression of its mitochondrial proteins or even increasing mitochondrial mass.

  • Line 263: "In a previous study where SIRT3 inhibition affected the glycolytic metabolism of a 263 cell,.. – which cell?
  • 263-271 – it is not clear where the authors are referring to their specific glioblastoma cell type, where some cells are from other studies, and where "cell " in general.

Response: We have addressed the concern regarding the clarity of the paragraph in lines 263-271. In response, we revised the paragraph to specify that the second part refers to glioblastoma cell lines. We hope that this clarification makes it evident that the discussion pertains specifically to our experimental context and aids in better understanding the flow of the manuscript.

We have changed the sentences to: In a previous study where SIRT3 inhibition affected the glycolytic metabolism of a cell, researchers found that the suppression of SIRT3 led to abnormal glycolysis in the kidney of diabetic mice [38]. In our results, we observe that probably, in response to the reduced function of SIRT3 due inhibition in both glioblastoma cell lines, the cells likely overexpress the affected mechanisms to counteract this effect.

  • Figure legends are indistinguishable from the main text.

Response: We acknowledge this concern and plan to address it by reducing the font size of the figure legends. By doing so, we aim to ensure that the figure legends are differentiated from the main text, facilitating easier identification for readers.

  • The title of the chapter 2.4. should be more specific. The current version is not informative, as it is well-known evidence that SIRT3 regulates the synthesis of proteins.

We have revised the subtitle to provide more informative content and changed it to Protein synthesis machinery is activated in response to SIRT3 inhibition, irrespective of cellular energy metabolism. 

Materials and Methods

  • The subsections are not numbered.

Response: We have now numbered the subsections to improve the organization and clarity of the document.

  • The tense must be kept consistent throughout the Manuscript but not vary from present to past (lines 253-254).

We have had the manuscript reviewed by a native English speaker to ensure grammatical accuracy and coherence. 

Discussion?

  • The MS lacks a Discussion section.

Response: Our results already have a discussion; however, we have added a discussion section to provide a comprehensive overview of the study's findings, their implications, and future research directions in the context of the broader literature on SIRT3 and cancer metabolism.

Conclusions

  • The section is, in part, presenting the content that could be fit for discussion. There are no clear conclusions stated.
  • Considering the claim about mitochondrial protein acetylation significance in glioblastoma tumours, the discussion of this aspect is insufficient.
  • The conclusions are not defined; the study's main finding is unclear.

Response: Thank you for your insightful comments regarding our manuscript. We appreciate the opportunity to address the points you raised and provide further clarity on the significance of our findings.

In response to your concerns about the discussion section, we acknowledge that the conclusions drawn from our study were not clearly articulated. We added a discussion section and modified our conclusions. We apologize for any confusion and have revised the discussion to provide more definitive conclusions based on our results.

Round 2

Reviewer 1 Report

Comments and Suggestions for Authors

No revisions were found. The author needs to provide track copy with changes marked. 

Comments on the Quality of English Language

Major revisions

Author Response

Reviewer 1

Comments:

No revisions were found. The author needs to provide track copy with changes marked.

Dear reviewer.

We regret that you have not found the file containing the change-tracked manuscript accessible. Unfortunately, the journal only gives the option of placing a file in the place of the manuscript, and in it, we have put the latest version with the accepted changes. which now contains the changes suggested by reviewer 1 in round 1 and the changes suggested by reviewer 2 in rounds 1 and 2 We will try to make available in the documents Manuscript_IJMS_20240226 with change tracking Version 1, which contains the changes suggested in round 1 by reviewers 1 and 2, and Manuscript_IJMS_20240226 with change tracking Version 2, which contains the changes suggested by reviewer 2 in round 2.

Reviewer 2 Report

Comments and Suggestions for Authors

The MS has been significantly improved; however, I still find that the Conclusions section should be more definitive. It is not enough to say that the differences are found between the acetylation profiles and SIRT-3 involvement in the metabolism of the two cell types because such a statement does not progress the reader much further from the initial state-of-the-art. The main findings should be better elaborated with regard to the title and main aims of the study, covering such questions as:

1-      What are the key differences between the regulatory pathways controlled by mitochondrial protein acetylation between the two metabolic GBM cell profiles?

2-      What are the key differences in the role of SIRT-3 in controlling the energetic metabolism of mitochondrial and glycolytic GBM cells?

Author Response

Reviewer 2

Comments:

The MS has been significantly improved; however, I still find that the Conclusions section should be more definitive. It is not enough to say that the differences are found between the acetylation profiles and SIRT-3 involvement in the metabolism of the two cell types because such a statement does not progress the reader much further from the initial state-of-the-art. The main findings should be better elaborated with regard to the title and main aims of the study, covering such questions as:

1-      What are the key differences between the regulatory pathways controlled by mitochondrial protein acetylation between the two metabolic GBM cell profiles?

2-      What are the key differences in the role of SIRT-3 in controlling the energetic metabolism of mitochondrial and glycolytic GBM cells?

Response:

We thank the Reviewer for these positive comments on our manuscript. We have addressed all the concerns raised in the revised version of the manuscript and pointed out important aspects that we must consider to fully comply with the objectives set by us when beginning this research.

In the new version of the manuscript, we have, from our point of view, covered the reviewer's request, substantially modifying the conclusion and fundamentally considering the two aspects indicated by the reviewer. We hope this new version meets the reviewer's quality standards.

Round 3

Reviewer 1 Report

Comments and Suggestions for Authors

Some of my comments have been addressed. In advance to some point, the introduction and discussion still lack a crucial understanding. This can be elaborated on in the introduction and discussion. It is insisted on elaborate and cite the following study:

i. Potential candidates from marine and terrestrial resources targeting mitochondrial inhibition: Insights from the molecular approach

ii. Mitochondrial defects in pancreatic beta-cell dysfunction and neurodegenerative diseases: Pathogenesis and therapeutic applications

Comments on the Quality of English Language

Minor revisions